# Impact of Endophthalmitis on the Risk of Acute Myocardial Infarction in Ankylosing Spondylitis Patients: A Population-Based Retrospective Cohort Study

**DOI:** 10.3390/jcm12031211

**Published:** 2023-02-03

**Authors:** Ting-Yi Lin, Yi-Fen Lai, Wu-Chien Chien, Yi-Hao Chen, Chi-Hsiang Chung, Jiann-Torng Chen, Ching-Long Chen

**Affiliations:** 1Department of Ophthalmology, Tri-Service General Hospital, National Defense Medical Center, Taipei City 11490, Taiwan; 2School of Public Health, National Defense Medical Center, Taipei City 11490, Taiwan; 3Department of Medical Research, Tri-Service General Hospital, National Defense Medical Center, Taipei City 11490, Taiwan; 4Graduate Institute of Life Sciences, National Defense Medical Center, Taipei City 11490, Taiwan; 5Taiwanese Injury Prevention and Safety Promotion Association, Taipei City 11490, Taiwan

**Keywords:** acute myocardial infarction 1, ankylosing spondylitis 2, endophthalmitis 3, infection 4, inflammation 5

## Abstract

Many studies have demonstrated an increased cardiovascular (CV) risk in ankylosing spondylitis (AS) patients. Nevertheless, the influence of an endophthalmitis episode toward the future risks of acute myocardial infarction (AMI) in AS patients has been unclear. The objective of this study was to explore the impact of endophthalmitis on AMI risk in this particular patient population by a population-based retrospective cohort study with a follow-up period up to 16 years. Univariate and multivariate Cox regression analyses were used for the risk evaluation and the results were presented as crude and adjusted hazard ratios (HRs). Overall, we enrolled 557 AS patients with endophthalmitis as the study cohort and selected another 2228 matched AS patients without endophthalmitis as the comparison cohort. Comparing the comparison cohort, the study cohort showed a significantly higher overall AMI incidence rate with an adjusted HR of 1.631 (*p* < 0.001). In conclusion, endophthalmitis increased the risk of AMI in AS patients after adjusting for possible clinical confounders. Special attention and work-up are required for physicians when encountering a history of endophthalmitis in these special patient populations, especially when they are comorbid with other potential CV risk factors.

## 1. Introduction

Ankylosing spondylitis (AS) is an axial spondyloarthritis (axSpA) that causes chronic inflammatory arthritis affecting the axial skeleton [1]. Typical clinical features of AS include inflammatory back pain, reduced spinal mobility, oligoarthritis (≤4 joints), enthesitis, and dactylitis [2]. Many studies have described an increased cardiovascular (CV) risk in AS patients [3,4,5]. An average of a 24% increase in the risk of myocardial infarction (MI) associated with AS was reported in a meta-analysis of 25 studies [6]. In addition, a multicenter cross-sectional study of 888 axSpA patients showed an increased CV risk in patients with extra-articular manifestations such as acute anterior uveitis, psoriasis, and inflammatory bowel disease [7]. Despite not fully understood, several possible underlying mechanisms were proposed to explain the increased CV risk in AS wherein inflammation is thought to play a key role. Inflammation not only has a direct effect but also influences patients’ CV risk factors, such as lipid profile, blood pressure, and insulin resistance [8].

Endophthalmitis is a severe form of ocular inflammation of both the anterior and posterior segments of the eye that can lead to irreversible visual loss mostly due to bacterial or fungal infections. Case reports regarding AS patients developing endophthalmitis during an episode of systemic infectious disease have been reported [9,10]. In addition, some cases of CV events, such as acute myocarditis or infective endocarditis, showed some association with endophthalmitis [11,12,13,14]. However, there are no studies investigating the influence of endophthalmitis on the future risk of acute myocardial infarction (AMI) in these autoimmune disease patients. In this study, we endeavored to evaluate the association between endophthalmitis and the risk of AMI in AS patients using a nationwide population-based database in Taiwan.

## 2. Materials and Methods

### 2.1. Data Source

The single-payer National Health Insurance Program was launched in March 1995 in Taiwan. More than 99% of Taiwan’s population (including foreigners) were enrolled in this program. The National Health Insurance Research Database (NHIRD) derived from this system contains registration files and original claim data for reimbursement. Based on the registration files and original claims data in the NHIRD, specific data subsets were constructed for research purposes. Longitudinal Health Insurance Database (LHID), which randomly selected 1,000,000 individuals from the Registry for Beneficiaries (ID) of the NHIRD, is a validated and representative data subset [15]. In our study, we used the Taiwan LHID database of 1,914,201 health-insured individuals to survey the impact of endophthalmitis on future AMI risk in the AS population from 1 January 2000 to 31 December 2015.

### 2.2. Ethical Considerations

This study was approved by the institutional review board of the Tri-Service General Hospital (TSGHIRB No.: E202216019). All the study procedures followed the tenets of the Declaration of Helsinki. Medical records in the NHIRD were scrambled before database construction and were further randomized before being released to the researchers. Therefore, it is theoretically impossible to identify individuals at any level using this database, so the need for informed patient consent was waived.

### 2.3. Patient Selection

The International Classification of Diseases, Ninth Revision, Clinical Modification (ICD-9-CM) codes were used for patient selection from the two-million-level Taiwan LHID from the year 2000 to 2015. Patients diagnosed with AS, defined as coding with ICD code 720.0 (ankylosing spondylitis) at least once during hospital admission or no less than three times during outpatient visits, were selected into the study population. Patients that met the inclusion criteria were excluded if they were diagnosed with AS before the inclusion date, AMI occurred before tracking, there was a lack of tracking medical data, age was less than 20 years, sex was unknown, medical records or insurance status were incomplete, or the given codes were unrelated or incorrect. Patients within the study population and coded with endophthalmitis (ICD codes 360.0, 360.00–360.04, and 360.1) in their medical records were further identified as the study cohort. After the study cohort were identified, a four-fold comparison cohort were chosen from AS patients without endophthalmitis after matching with factors including sex, age, patient comorbidities, and the inclusion date. For all the study participants, the follow-up end points were the incidence of AMI (ICD code 410), withdrawal from the National Health Insurance Program, or tracking endpoint of the study (31 December 2015).

### 2.4. Comorbidities

Patient comorbidities were recorded at baseline and at the study endpoint for further analysis. The diagnosis of a comorbidity was defined as receiving the same diagnosis in three or more medical visits within a year before the inclusion date or the date of the study endpoint. The Charlson comorbidity index revised score (CCI_R), an assessment tool designed specifically to predict long-term mortality risk, was calculated as well. The algorithms using ICD-9-CM codes to identify patients with AS, endophthalmitis, AMI and comorbidities were listed in Appendix A.

### 2.5. Statistical Analyses

The results of continuous variables are presented as mean ± standard deviation, and the results of categorical variables are displayed as numbers and percentages. We used an independent Student *t*-test to determine the statistical difference between continuous variables. Furthermore, we assessed the differences in categorical variables by Pearson’s chi-square test or Fisher’s exact tests. Meanwhile, the hazard ratios (HRs) for the association between clinical variables and related comorbidities with AMI were evaluated using univariate and multivariate Cox regression analyses. We then stratified patients into different subgroups based on relevant clinical variables and patient’s comorbidities. Further analyses between these subgroups were carried out. Kaplan–Meier survival curves and log-rank tests were used to evaluate the cumulative risk of AMI in AS patients with and without endophthalmitis in this cohort. Statistical significance was defined as *p* < 0.05. All statistical analyses were performed using IBM Statistical Package for the Social Sciences (SPSS) Statistics for Windows, version 22.0. (IBM Corp. Released 2013. Armonk, NY, USA).

## 3. Results

From 2000 to 2015, we identified 21,846 AS patients from the 1,914,201 registered individuals of Taiwan LHID. A total of 1850 patients were excluded due to reasons showed in the exclusion criteria, leaving a final study population of 19,996 AS patients. Within the study population, 557 patients with endophthalmitis were enrolled in the study cohort. Subsequently, 2228 AS patients without endophthalmitis matched with sex, age, comorbidities, and inclusion date with the study cohort were selected as the comparison cohort. Figure 1 shows the flowchart of the process by which the study population was selected. At baseline, the two cohorts included 55.12% men and 44.88% women with an overall mean age of 37.72 ± 18.92 years. Moreover, there was no statistical difference in the baseline demographic and medical characteristics such as sex, age, patient comorbidities, and CCI_R score between the two cohorts.

At the end of this study, 69 (12.39%) and 193 (8.66%) patients had AMI in the study cohort and the comparison cohort, respectively (*p* < 0.001). The mean ages of the study cohort and the comparison cohort were 40.53 ± 19.18 years and 41.33 ± 19.90 years, respectively. In addition, patients in the study cohort had a higher proportion of DM, hyperlipidemia, HTN, CHF, COPD, asthma, and higher CCI_R scores compared with the comparison cohort (see Table 1). The comorbidity bar graphs of the enrolled patients at baseline and endpoint are shown in Figure 2. Patients were followed up for an average of 9.91 ± 8.57 years, with no difference between AS patients with and without endophthalmitis (Appendix A).

Figure 3 shows the Kaplan–Meier survival curves for the cumulative risk of AMI in the two cohorts. As the figure showed, the cumulative AMI risk was significantly higher in the study cohort than the comparison cohort (*p* < 0.001; log-rank test). Furthermore, the median time from the beginning of follow up to the AMI incidence of participants were 1.92 and 2.24 years in the study cohort and the comparison cohort, respectively (*p* < 0.001; Mann–Whitney U test; shown in Appendix A).

The risk of AMI analyzed by Cox regression is summarized in Table 2. The results show that the risk of AMI was significantly higher in the study cohort than in the comparison cohort. Compared with the reference value, it can be seen that the risk of AMI in AS patients aged 40–59 years and ≥60 years is significantly increased; in addition, patients with other underlying comorbidities, such as DM, hyperlipidemia, HTN, CVA, CHF, COPD, asthma, and CAD, and AS patients who had higher CCI_R scores also had increased risk of AMI.

All of the statistical significance remained after adjusting for all listed covariates. Figure 4 shows the crude and adjusted HRs of the clinical variables analyzed in Table 2 with forest plots.

We further stratified the two cohorts using the listed clinical variables in Table 3. Analysis of AMI risk between these stratified subgroups were then conducted and presented. AS patients with/without endophthalmitis had an overall AMI incidence rate (IR) of 1278.08 and 872.93 per 10^5^ person years (PYs), respectively (adjusted HR = 1.631, *p* < 0.001). Among all the stratified subgroups, the adjusted HR was significantly increased in the study cohort compared with the comparison cohort (*p* < 0.001 in all subgroups). In light of our study results, AS patients with underlying comorbidities concurrently resulted in an even higher adjusted HRs of AMI than those without comorbidities.

## 4. Discussion

To the best of our knowledge, this is the first study to investigate the association of endophthalmitis and AMI risk in AS patients. According to the results of Cox regression, in the study population included in this study, endophthalmitis, 40–59 years old and >60 years old, DM, hyperlipidemia, HTN, CVA, CHF, COPD, asthma, CAD and higher CCI_R scores were considered as risk factors for AMI. AS patients with endophthalmitis showed a significant elevation in the AMI risk compared with AS patients without endophthalmitis (adjusted HR = 1.631; *p* < 0.001). The statistical significance remained irrespective of sex, age, or clinical comorbidities such as DM, hyperlipidemia, HTN, CVA, CHF, COPD, asthma, and CAD, indicating that endophthalmitis is an independent risk factor for AMI in AS patients. In addition, the adjusted HRs for AMI were higher in AS patients with comorbidities concurrently. Likewise, AS patients with endophthalmitis presented a significantly greater cumulative AMI risk than those without endophthalmitis by Kaplan–Meier analysis.

Our study included two cohorts matched with factors including sex, age, comorbidities, and inclusion date. At the study endpoint, significantly higher levels of DM, hyperlipidemia, HTN, CHF, COPD, and asthma were noticed in the cohort of AS patients with endophthalmitis. Despite many studies investigating preoperative, perioperative, and postoperative factors of endophthalmitis, no previous study has investigated the relationship between endophthalmitis episodes and the future incidence of comorbidities in AS patients. Furthermore, there was no difference in the all-cause mortality rates between the two cohorts. However, an increased mortality rate was found in endophthalmitis patients concurrently having renal disease, septicemia, pneumonia, tumors, and AS in our previous studies [16,17]. This controversial finding may be due to the retrospective study design and relatively small sample size of our database studies. As a result, the contribution of these factors to patient mortality must be further evaluated with a well-designed, prospective cohort study.

As shown in Table 2, some statistically significant risk factors for AMI in our study population were identified. As expected, AS patients with traditional risk factors for CV events such as DM, hyperlipidemia, HTN, and older age resulted in a higher risk of AMI [18,19]. Furthermore, age, severity of autoimmune disease, male sex, HTN, DM, and hyperlipidemia are also some of the most commonly found risk factors in studies of atherosclerosis in patients with autoimmune rheumatic diseases [20]. A Swedish study investigating the first episode of AMI in AS patients showed a higher comorbidity burden including stable ischemic heart disease, CHF, pulmonary diseases, DM, HTN, and renal disease than that in the general population [21].

As shown in Table 3, the overall IRs of AMI were 1278.08 and 872.93 per 10^5^ PYs in AS patients with and without endophthalmitis, respectively. The IR of AMI was reported to be 50.7 per 10^5^ PYs in 2015 in Taiwan, which is much less than that in the AS population in our study [22]. This is consistent with previous study results that showed increased IRs of AMI in AS patients [23,24]. However, our study population was confined to newly diagnosed AS patients aged ≥20 years who had a complete medical record in the database, and the IRs were not age- and sex-adjusted in this study. Consequently, the epidemiological findings in our study required further research to clarify and need to be interpreted with caution.

In the present study, endophthalmitis was an independent risk factor for AMI in AS patients. Previous studies have revealed an association between acute infections and an increased risk of AMI in the short- to long-term period [25]. In a prospective AMI survey of 4573 patients, post-infectious AMI was common, accounting for 10% of all cases, and doubled in-hospital mortality [26]. The exact pathogenesis of how infectious diseases increase AMI risk is not fully understood, but inflammation plays an important role. Several potential mechanisms have been proposed such as increased circulating inflammatory cytokines, activation of inflammatory activity in atheromatous plaques, upregulation of host response proteins, production of neutrophil extracellular trap, increased platelet activity, increased generation of procoagulants, impaired fibrinolysis, endothelial dysfunction, increased metabolic demands, toxin-mediated vasoconstriction, infection-induced cardiac lesions, and cytokine storm [25]. Both systemic and local levels of the cytokines have been noticed to increase in either uveitis with infectious or non-infectious etiologies [27,28,29]. Moreover, cytokines such as interleukin (IL)-1, IL-6, IL-8, and tumor necrosis factor α, which can activate inflammatory cells in atherosclerotic plaque, have been found to be elevated during endophthalmitis [25,30,31]. Acute infections, such as endophthalmitis, have been proposed to accelerate and exacerbate the condition of cardiovascular diseases through increased systemic inflammatory responses. This condition may be even more prominent in patients with underlying chronic inflammatory diseases such as AS. However, there is a scarcity of literature investigating the systemic response of endophthalmitis. Further research is needed to explore systemic inflammation during endophthalmitis episodes and its influence on AMI.

This study has several strengths. First, this is the first study to investigate the risk association between endophthalmitis and AMI in AS patients. Second, the Taiwan LHID, a validated and representative nationwide million-level database, was used in our study. In addition, a long follow-up period of up to 16 years and the two-million-level sample size of our study could theoretically provide high statistical power of the study results. Third, the influence of possible confounders was diminished by using univariate and multivariate Cox regression analyses.

However, some limitations also existed in our study. First, this is a retrospective study using medical records from a nationwide database. The causal relationship between endophthalmitis and incidence of AMI in AS patients is difficult to determine, owing to the study design. Second, our study population mostly included Taiwanese people. The disease prevalence may vary between nations and races; therefore, the generalizability of our study results in other populations or countries need further confirmation. Third, data regarding disease activity and severity of AS, endophthalmitis, patient morbidities, and therapeutic medications prescribed could not be obtained from LHID. As a result, potential biases caused by these factors during the cohort could not be justified. Fourth, factors including different etiologies, infectious pathogens, and endogenous or exogenous types of endophthalmitis were not further analyzed in our study. Fifth, time-dependent covariates were not applied in the Cox regression analyses. Finally, the results of laboratory examinations and imaging were also unavailable in the NHIRD.

## 5. Conclusions

In conclusion, endophthalmitis increased the risk of AMI in AS patients after adjusting for possible clinical confounders. Based on our study results, special attention and work-up are required for physicians when encountering a history of endophthalmitis in these special patient populations, especially when they are comorbid with other potential CV risk factors.

## Figures and Tables

**Figure 1 jcm-12-01211-f001:**
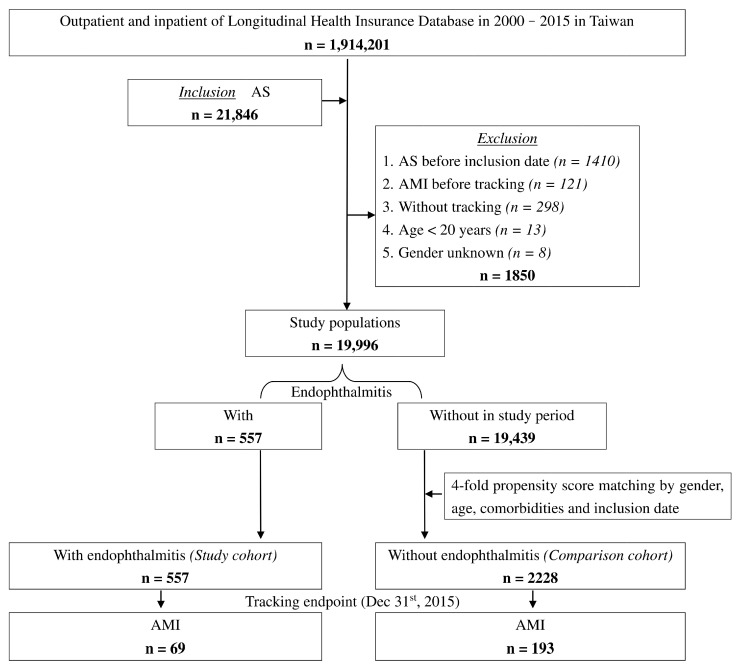
Flowchart of the patient selection in this cohort.

**Figure 2 jcm-12-01211-f002:**
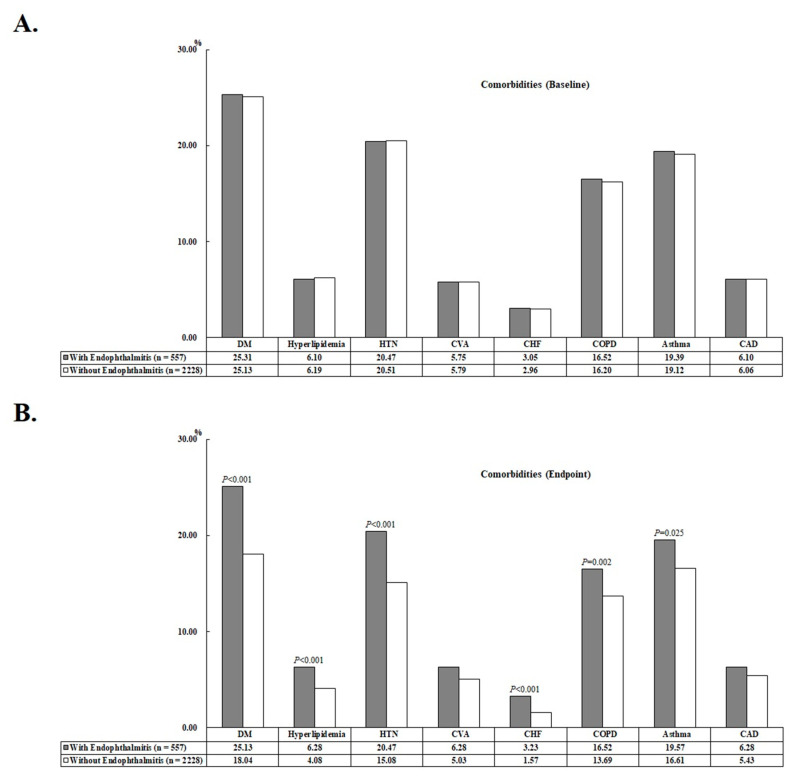
Bar chart of the comorbidities of the enrolled patients at baseline (**A**) and at the study endpoint (**B**).

**Figure 3 jcm-12-01211-f003:**
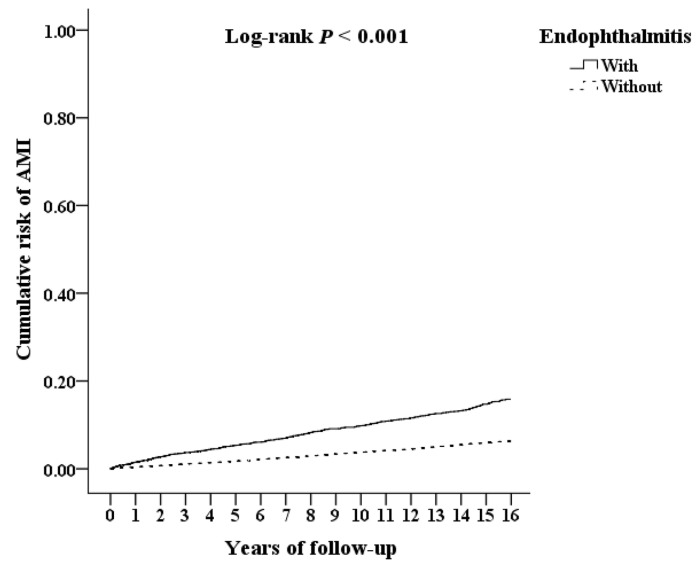
Kaplan–Meier survival curve of the cumulative risk of AMI in AS patients with and without endophthalmitis.

**Figure 4 jcm-12-01211-f004:**
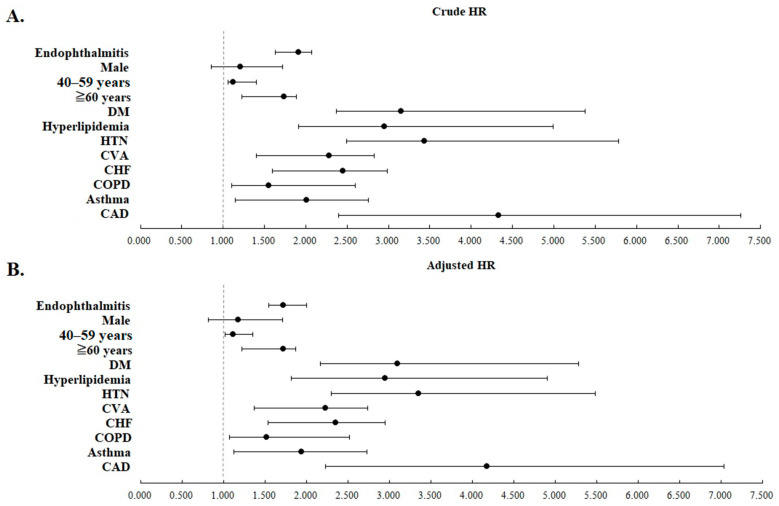
Forest plots of the crude (**A**) and adjusted (**B**) HRs of clinical variables analyzed in Table 2. All variables listed in Table 2 were adjusted in adjusted HR.

**Table 1 jcm-12-01211-t001:** Demographic and medical characteristics of enrolled AS patients with and without endophthalmitis at the endpoint of the study.

Endophthalmitis	Total		With		Without		*p* Value
Characteristics	*n*	%	*n*	%	*n*	%	
Total	2785		557	20.00	2228	80.00	
AMI	262	9.41	69	12.39	193	8.66	<0.001
Gender							0.999
Male	1535	55.12	307	55.12	1228	55.12	
Female	1250	44.88	250	44.88	1000	44.88	
Age (years)	41.18 ± 19.77		40.53 ± 19.18		41.33 ± 19.90		0.045
Age group (years)							0.086
20–39	1619	58.13	308	55.30	1311	58.84	
40–59	770	27.65	154	27.65	616	27.65	
≧60	396	14.22	95	17.06	301	13.51	
DM	542	19.46	140	25.13	402	18.04	<0.001
Hyperlipidemia	126	4.52	35	6.28	91	4.08	<0.001
HTN	450	16.16	114	20.47	336	15.08	<0.001
CVA	147	5.28	35	6.28	112	5.03	0.145
CHF	53	1.90	18	3.23	35	1.57	<0.001
COPD	397	14.25	92	16.52	305	13.69	0.002
Asthma	479	17.20	109	19.57	370	16.61	0.025
CAD	156	5.60	35	6.28	121	5.43	0.368
CCI_R	0.91 ± 1.11		0.99 ± 1.14		0.89 ± 1.10		<0.001
All-cause mortality	222	7.97	49	8.80	173	7.76	0.324

AMI, acute myocardial infarction; CAD, coronary artery disease; CCI_R, Charlson comorbidity index revised; CHF, congestive heart failure; COPD, chronic obstructive pulmonary disease; CVA, cerebrovascular accident; DM, diabetes mellitus; HTN, hypertension.

**Table 2 jcm-12-01211-t002:** AMI risk evaluation by Cox regression analysis.

Variables	Crude HR (95% CI)	*p*	Adjusted HR (95% CI)	*p*
Endophthalmitis				
Without	Reference		Reference	
With	1.821 (1.527–2.027)	<0.001	1.631 (1.464–1.898)	<0.001
Gender				
Male	1.169 (0.790–1.565)	0.302	1.144 (0.766–1.528)	0.309
Female	Reference		Reference	
Age (yrs)				
20–39	Reference		Reference	
40–59	1.106 (1.043–1.308)	0.028	1.074 (1.003–1.266)	0.047
≧60	1.617 (1.205–1.764)	<0.001	1.511 (1.144–1.736)	<0.001
DM				
Without	Reference		Reference	
With	2.962 (2.041–4.993)	<0.001	2.947 (2.023–4.937)	<0.001
Hyperlipidemia				
Without	Reference		Reference	
With	2.714 (1.560–4.312)	<0.001	2.586 (1.508–4.203)	<0.001
HTN				
Without	Reference		Reference	
With	3.155 (2.069–5.099)	<0.001	3.063 (2.020–5.088)	<0.001
CVA				
Without	Reference		Reference	
With	2.225 (1.366–2.734)	<0.001	2.137 (1.244–2.652)	<0.001
CHF				
Without	Reference		Reference	
With	2.349 (1.532–2.948)	<0.001	2.329 (1.470–2.856)	<0.001
COPD				
Without	Reference		Reference	
With	1.479 (1.054–2.448)	<0.001	1.448 (1.039–2.377)	0.003
Asthma				
Without	Reference		Reference	
With	1.930 (1.075–2.659)	<0.001	1.919 (1.041–2.606)	0.009
CAD				
Without	Reference		Reference	
With	4.188(2.289–7.094)	<0.001	4.090 (2.191–7.017)	<0.001
CCI_R	1.210 (1.144–1.226)	<0.001	1.176 (1.132–1.210)	<0.001

All variables listed in the table were adjusted in adjusted HR. AMI, Acute myocardial infarction; CAD, coronary artery disease; CCI_R, Charlson comorbidity index revised; CHF, congestive heart failure; COPD, chronic obstructive pulmonary disease; CVA, cerebrovascular accident; DM, diabetes mellitus; HTN, hypertension.

**Table 3 jcm-12-01211-t003:** Risk analysis of AMI between patients with and without endophthalmitis in subgroups stratified with clinical variables.

Endophthalmitis	With		Without (Reference)		Adjusted HR (95% CI)	*p*
Stratified Subgroups	Events	Rate (per 10^5^ PYs)	Events	Rate (per 10^5^ PYs)		
Total	69	1278.08	193	872.93	1.631 (1.464–1.898)	<0.001
Gender						
Male	37	1242.71	102	837.40	1.680 (1.509–1.956)	<0.001
Female	32	1321.56	91	916.51	1.602 (1.437–1.863)	<0.001
Age (yrs)						
20–39	39	1308.56	126	968.74	1.534 (1.386–1.796)	<0.001
40–59	19	1278.41	55	897.82	1.571 (1.411–1.828)	<0.001
≧60	11	1180.07	12	403.10	3.227 (2.896–3.753)	<0.001
DM						
Without	55	1361.57	181	996.68	1.544 (1.386–1.797)	<0.001
With	14	1029.94	12	303.86	3.272 (2.938–3.808)	<0.001
Hyperlipidemia						
Without	62	1223.69	186	878.26	1.575 (1.415–1.833)	<0.001
With	7	2107.86	7	751.68	3.089 (2.773–3.594)	<0.001
HTN						
Without	52	1210.93	179	954.19	1.432 (1.286–1.667)	<0.001
With	17	1539.13	14	417.88	4.148 (3.724–4.826)	<0.001
CVA						
Without	65	1276.32	187	893.15	1.596 (1.433–1.858)	<0.001
With	4	1307.27	6	511.81	3.282 (2.946–3.818)	<0.001
CHF						
Without	67	1277.94	192	880.60	1.653 (1.483–1.922)	<0.001
With	2	1282.63	1	326.50	3.246 (2.915–3.778)	<0.001
COPD						
Without	64	1415.78	188	981.41	1.620 (1.455–1.885)	<0.001
With	5	569.31	5	169.29	3.343 (3.001–3.890)	<0.001
Asthma						
Without	62	1417.35	185	1014.80	1.591 (1.428–1.851)	<0.001
With	7	683.34	8	206.22	3.272 (2.937–3.808)	<0.001
CAD						
Without	64	1268.26	189	905.25	1.570 (1.410–1.827)	<0.001
With	5	1418.60	4	324.85	5.165 (4.637–6.010)	<0.001

All variables listed in the table were adjusted in adjusted HR. AMI, Acute myocardial infarction; CAD, coronary artery disease; CHF, congestive heart failure; COPD, chronic obstructive pulmonary disease; CVA, cerebrovascular accident; DM, diabetes mellitus; HTN, hypertension.

## Data Availability

The original contributions presented in the study are included in the article/Appendix A. Further inquiries can be directed to the corresponding author.

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
