# Peer review of "Impact of Endophthalmitis on the Risk of Acute Myocardial Infarction in Ankylosing Spondylitis Patients: A Population-Based Retrospective Cohort Study"

_jcm, 2023, doi:10.3390/jcm12031211_

Round 1

Reviewer 1 Report

I have the following comments to be addressed:

- This citation (PMID: 36150319, "Cardiovascular and disease-related features associated with extra-articular manifestations in axial spondyloarthritis. A multicenter study of 888 patients) should be quoted in the Introduction.

- I do not think that the asterisks are necessary in Table 1. It is already denoted, by the p-values in the table, which is less than 0.05 and 0.001. This also applies for other tables.

- Figure 3 should be displayed with the Y axis up to 1 and also as-is. Doing it with the Y axis up to 1 would help to know what the total real incidence is like and not only the differences between the survival lines.

- Lines 138-140 must specify exactly what 'years' refers to, median number of years? By what statistical method has it been measured?

- What variables were selected to make the adjustment in Figure 4? The variables that were adjusted should be described at the bottom of the figure or in the text.

- No reference is made, I believe, in the manuscript to the therapies used at the beginning of the study and during follow-up (cumulative doses, anti-TNF, etc.). This information is of interest and could justify bias if it has not been collected. If this has been the case, it should be mentioned as a limitation of the study.

The fact of not including, in the models of reg. de Cox, time dependent variables, should also be mentioned as a limitation.

- Also missing is the fact that it is not mentioned whether patients with endophthalmitis have a more severe/active disease. If this were the case, which the authors do not mention, one might think that it is not the endophthalmitis that causes the greatest risk of AMI, but rather the severity/activity of the disease itself. This must be justified.

Reviewer 2 Report

In the article to be reviewed, the primary end point is the occurrence of a myocardial infarction defined by the corresponding International Classification of Diseases code: "For all the study participants, the follow-up end points were the incidence of Acute Myocardial Infarction (ICD 87 code 410)".

In the paper already published, the primary end point is the occurrence of acute coronary syndrome, which they defined as a composite end point and which corresponds to both the myocardial infarction and unstable angina codes: "All patients were followed from the index date until the incidence of ACS (ICD-9-CM code of AMI: 410, and ICD-9-CM code of unstable angina: 411.1, 411.8).

Thus the criterion "myocardial infarction" is a sub-criterion of the criterion "acute coronary syndrome" used in the first study.

Furthermore: in the study already published, there were 283 events (i.e. 283 acute coronary syndromes) with 83 ACS out of 530 patients in the group with endophthalmitis and 200 ACS out of 2120 patients in the group without endophthalmitis. In the review study, there were 262 events (i.e. 262 myocardial infarctions) with 69 infarctions out of 557 patients in the group with endophthalmitis and 193 infarctions out of 2228 patients in the group without endophthalmitis. It can therefore be seen that the majority of ACS events in the first study were therefore myocardial infarctions. It is therefore not surprising that the results are very similar in the two studies because the endpoint is almost the same.

Round 2

Reviewer 1 Report

Point 4. It is well established that time in Cox regression models cannot be expressed as mean/SD due to the presence of censored data. This must be corrected and adequately expressed perhaps by non-parametric method or median/quartiles.
